# Forest Fragmentation and Fires in the Eastern Brazilian Amazon–Maranhão State, Brazil

Celso H. L. Silva-Junior [1,2,3,*,†], Arisson T. M. Buna [4,†], Denilson S. Bezerra [5,†], Ozeas S. Costa, Jr. [6], Adriano L. Santos [5], Lidielze O. D. Basson [5], André L. S. Santos [7], Swanni T. Alvarado [3], Catherine T. Almeida [8], Ana T. G. Freire [5], Guillaume X. Rousseau [3], Danielle Celentano [3,9], Fabricio B. Silva [4], Maria S. S. Pinheiro [5], Silvana Amaral [10], Milton Kampel [10], Laura B. Vedovato [11], Liana O. Anderson [12] and Luiz E. O. C. Aragão [10,11]

1. Institute of Environment and Sustainability, University of California, Los Angeles, CA 90095, USA
2. Jet Propulsion Laboratory, California Institute of Technology, Pasadena, CA 91011, USA
3. Departamento de Engenharia Agrícola, Universidade Estadual do Maranhão—UEMA, São Luís 65000-000, MA, Brazil; swanni_ta@yahoo.es (S.T.A.); guilirous@yahoo.ca (G.X.R.); danicelentano@yahoo.com.br (D.C.)
4. Mestrado em Meio Ambiente, Universidade CEUMA—UniCEUMA, São Luís 65075-120, MA, Brazil; arisson.buna@ceuma.br (A.T.M.B.); fabricio.brito@ceuma.br (F.B.S.)
5. Departamento de Oceanografia e Limnologia, Universidade Federal do Maranhão—UFMA, São Luís 65080-805, MA, Brazil; denilson.bezerra@ufma.br (D.S.B.); adriano.ls@discente.ufma.br (A.L.S.); lidielze.dourado@discente.ufma.br (L.O.D.B.); ana.talita@discente.ufma.br (A.T.G.F.); pinheiro.socorro@ufma.br (M.S.S.P.)
6. School of Earth Sciences, The Ohio State University, Mansfield, OH 43210, USA; costa.47@osu.edu
7. Instituto Federal do Maranhão—IFMA, São Luís 65030-005, MA, Brazil; andresantos@ifma.edu.br
8. Escola Superior de Agricultura Luiz de Queiroz, Universidade de São Paulo—USP, Piracicaba 13418-900, SP, Brazil; catherine.almeida@usp.br
9. Conservação Internacional—CI, Brasília 70711-902, DF, Brazil
10. Instituto Nacional de Pesquisas Espaciais—INPE, São José dos Campos 12227-010, SP, Brazil; silvana.amaral@inpe.br (S.A.); milton.kampel@inpe.br (M.K.); luiz.aragao@inpe.br (L.E.O.C.A.)
11. College of Life and Environmental Sciences, Geography, University of Exeter, Exeter EX4 4PY, UK; lv287@exeter.ac.uk
12. Centro Nacional de Monitoramento e Alertas de Desastres Nacionais—CEMADEN, São José dos Campos 12630-000, SP, Brazil; liana.anderson@cemaden.gov.br
* Correspondence: celsohlsj@gmail.com
† These authors contributed equally to this work.

**Abstract:** Tropical forests provide essential environmental services to human well-being. In the world, Brazil has the largest continuous area of these forests. However, in the state of Maranhão, in the eastern Amazon, only 24% of the original forest cover remains. We integrated and analyzed active fires, burned area, land use and land cover, rainfall, and surface temperature datasets to understand forest fragmentation and forest fire dynamics from a remote sensing approach. We found that forest cover in the Maranhão Amazon region had a net reduction of 31,302 km$^2$ between 1985 and 2017, with 63% of losses occurring in forest core areas. Forest edges extent was reduced by 38%, while the size of isolated forest patches increased by 239%. Forest fires impacted, on average, around $1031 \pm 695$ km$^2$ year$^{-1}$ of forest edges between 2003 and 2017, the equivalent of 60% of the total burned forest in this period. Our results demonstrated that forest fragmentation is an important factor controlling temporal and spatial variability of forest fires in the eastern Amazon region. Thus, both directly and indirectly, forest fragmentation can compromise biodiversity and carbon stocks in this Amazon region.

**Keywords:** deforestation; Maranhão; climate change; carbon emissions; land conversion

## 1. Introduction

Tropical forests provide essential environmental services to human well-being, including the storage of biodiversity and carbon [1–4]. The Amazon rainforest is the largest tropical forest globally, covering almost 7 million square kilometers, and extending into nine countries in South America (Brazil, Peru, Colombia, Venezuela, Ecuador, Bolivia, Guyana, Suriname, and French Guiana) [5]. Brazil holds the largest continuous area of forest in the region—around 60% of the Amazon basin [5]—distributed over nine federative units. One of these units, the state of Maranhão, had an original old-growth forest cover of 110,400 km$^2$ (around 33% of the state's territory) [6]. This region is part of the Belém Endemism Center, where around 30 species of animals and plants are on the endangered species list [7]. Large-scale deforestation since the mid-1960s—primarily for agriculture and cattle ranching—has resulted in the loss of a significant portion of the original forest cover in the Maranhão Amazon [6,8], and only around 23,967 km$^2$ (24%) remains (up until 2019) [9]. Maranhão has the second-highest level of forest fragmentation among all states in the Brazilian Legal Amazon, with almost 74% of the area represented in the non-core fragmentation category [10].

Deforestation alters the spatial configuration of forest cover through fragmentation, increasing the edge areas, and reducing the connectivity of the core remnants [11–13]. Among other negative impacts [12,14], fragmentation makes the forest more susceptible to fire due to the edge effects [13,15,16]. Forest edges and small forest patches have suitable conditions for the spread of fire, including a drier and hotter microclimate, abundant availability of fuel (necromass) in the understory, and greater exposure to ignition sources, since they are surrounded by agricultural and livestock areas [13,15,17–20].

Although the fire dynamics are well-documented in other biomes of Maranhão [21–25], little is known about the forest fires and their relationship with forest fragmentation in the Amazon biome of the state. Moreover, extreme droughts combined with anthropogenic ignition sources [26], such as land management fires, deforestation fires, and arson fires, have caused fires to spread uncontrollably through the Amazon forests [6,27–30]. This synergy between highly fragmented forests, abundant ignition sources, and the frequent occurrence of extreme drought can lead to forest fires that result in respiratory illness in the population [31,32], loss of biodiversity [33], and carbon stocks [34–36] beyond the commitment of other environmental services [37] of the Amazon Forest remnants in Maranhão.

Thus, a better understanding of the trajectory of forest fragmentation, the dynamics of forest fires, and their interactions is crucial for providing decision-makers with the tools needed for the elaboration of policies and enforcement strategies aimed at managing fire prevention and forest fragmentation in the Maranhão Amazon region. This study contributes to such understanding by identifying the levels of forest fragmentation in the Maranhão Amazon between 1985 and 2017, providing an explicit forest fires map in the region between 2003 and 2017, and exploring the relationship between fragmentation and forest fires during this period. This allows decision-makers to define priority areas for fire management and ecosystem restoration.

## 2. Materials and Methods

### 2.1. Study Area

The Maranhão state is located in the northeast region of Brazil, has a territorial area of 329,651.495 km$^2$, and an estimated population in 2021 of 7,153,262 inhabitants [38]. The state straddles the transition between the Amazon and Cerrado biomes [21,22,39], with the Amazon occupying an area of 115,139 km$^2$, or 35% of the state's territory (Figure 1) [40].

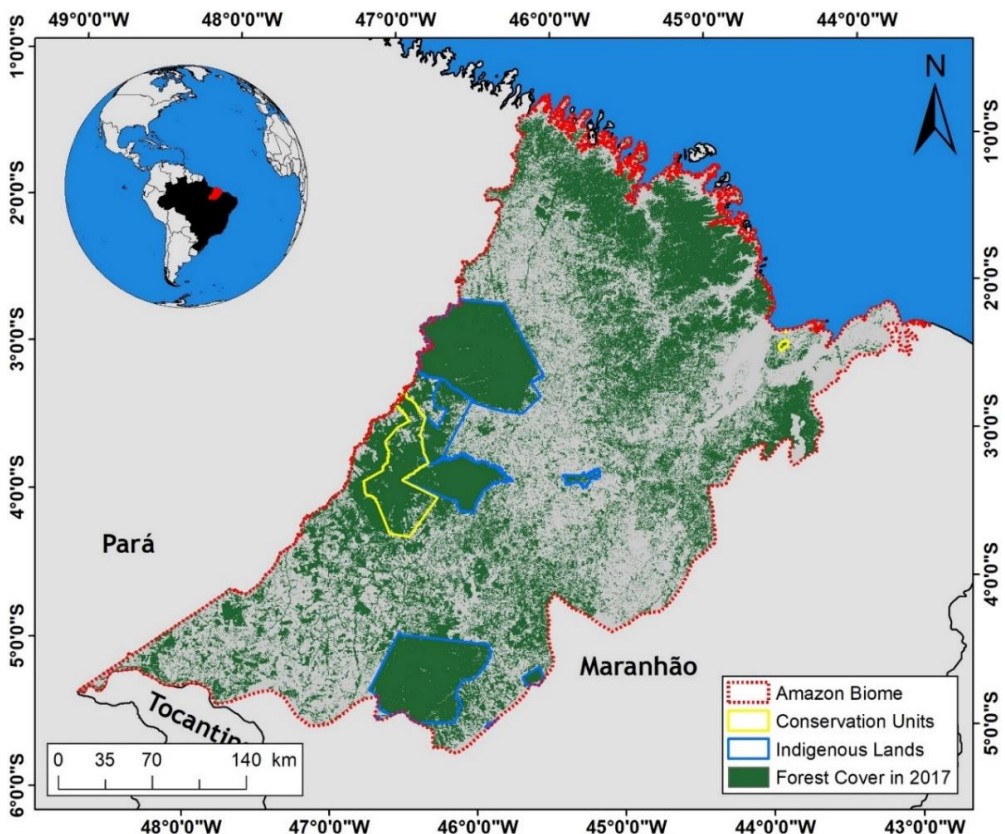

**Figure 1.** Map of the study area (the Amazon biome in Maranhão state). Forest cover from the MapBiomas project [41].

Of the remaining old-growth forest cover (24,700 km$^2$) inside Maranhão's Amazon region in 2016, around 70% exists in Conservation Units and Indigenous Lands, 10% in agrarian reform settlements, and another 20% scattered on private rural properties [6]. Between 2007 and 2015, degraded forests in the "Mosaico Gurupi" (the region where the most significant proportion of the forest remnants of the Maranhão Amazon region is located) totaled 2200 km$^2$, or the equivalent of 14% of remaining forests in protected areas in the region [8]. The Arariboia Indigenous Land, for example, had 45% of its mature forests degraded in this period [8].

### 2.2. Dataset

A total of five remote sensing datasets were used in this study, including land use and land cover, active fires, burned areas (fire scars), rainfall, and surface temperature.

### 2.2.1. Land Use and Land Cover Data

Land use and land cover data were obtained from the MapBiomas Project [41], Collection 3. Annual maps were retrieved for the period between 1985 and 2017. This dataset was obtained by classifying Landsat satellite images with 30-m spatial resolution [42,43], using a theoretical algorithm implemented in a planetary-scale geospatial analysis cloud platform [44]. The data have an overall accuracy of 94.80%, allocation disagreement with 2.80%, and area disagreement with 2.40% [45,46].

### 2.2.2. Active Fires Data

To understand the forest fire activity in our study area, we used the VNP14IMGTDL_NRT product. As the VNP14IMGTDL_NRT product started in 2012, active fire data from the NASA VIIRS were obtained for the 2012–2017 period. This product is generated from information provided by the VIIRS (Visible Infrared Imaging Radiometer Suite) sensor onboard



the NASA/NOAA satellite Suomi National Orbiting Polar Partnership (Suomi NPP) and the JPSS Series satellite NOAA-20 [47,48]. These data have a nominal spatial resolution of 375 m. The VIIRS active fire data provide a more significant response to fires in relatively small areas and improved performance in nighttime detection [47,49]. Consequently, these data are suitable for supporting fire management and scientific applications that require improved detection fidelity [47,49]. In addition, these data are a proxy for the fire activity on the land surface.

### 2.2.3. Burned Areas Data

To understand the area (extent) impacted by forest fires in our study area, we used the MODIS MCD64A1 product. Monthly burned areas data, or fire scars, were obtained for the 2003–2017 period from the MODIS MCD64A1 product, Collection 6 [50]. The mapping approach of this product employs the contextual analysis of MODIS sensor surface reflectance images in 500 m nominal spatial resolution, plus detections of active fires data with 1 km spatial resolution, also derived from the MODIS sensor [51]. Global and local validations (for the Amazon and Cerrado biomes) showed a suitable performance for this product [52–56].

### 2.2.4. Rainfall Precipitation Data

Monthly rainfall data for the 2012–2017 period (the same period as active fires data) were obtained from the Climate Hazards Group InfraRed Precipitation with Station data (CHIRPS), with 0.05° of spatial resolution [57]. The CHIRPS rainfall dataset integrates satellite imagery with rain gauge station data to create a time series in a regular pixel grid [57]. In the Brazilian Amazon, validations have shown that the product explains around 73% of the rainfall measured in the field by rain gauges, with a root mean square error below 15 mm month$^{-1}$ [58].

### 2.2.5. Land Surface Temperature Data

Monthly surface temperature data were obtained for the 2012–2017 period (the same period as active fires data) using the MODIS MOD11C3 product, Collection 6, available with a spatial resolution of 0.05°. The resulting monthly dataset was derived by compositing and averaging the daily temperature values from the corresponding month of the MODIS MOD11C1 product [59]. The calibration of the product is periodic and performed through data collected in the field, as well as studies of surface radiance [59].

### *2.3. Data Processing and Analysis*

### 2.3.1. Forest Fragmentation Analysis

We first reclassified the original MapBiomas land use and land cover maps to the binary maps. Then, we consider the class "Forest Formation" (including old- and second-growth forests) as a forest cover in our analyses. Therefore, we assigned the value 1 for forest pixels, and 0 for other land cover pixels.

Then, the binary maps were classified using the Morphological Segmentation of Binary Patterns—MSPA algorithm [60], implemented using GUIDOS, a free software toolbox dedicated to quantitative analysis of digital landscape images (Available online: https://forest.jrc.ec.europa.eu/en/activities/lpa/gtb, accessed on 20 January 2022).

The MSPA is a customized sequence of mathematical morphological operators targeted at describing the geometry and connectivity of the image components [60]. The MSPA's approach is based on geometric concepts, and can be applied on any scale. The foreground area of a binary image is divided into the following visually distinguished MSPA classes: *Edge*—the outer perimeter of the forest class (including the Loop, Corridor, and Branch classes); *Perforation*—edges inside core areas; *Core*—the forest core area, excluding the edges; and *Islets*—the forest sections with an area too small to contain a core. More details and illustrations are available in the MSPA Guide (https://forest.jrc.ec.europa.eu/en/activities/

lpa/mspa, accessed on 20 January 2022). An example of the MSPA classification is shown in Figure A1.

The edge width of 1020 m (or 34 pixels, due to the spatial resolution of the forest cover data) was used to include the main edge effects, such as canopy desiccation, fire susceptibility, wind turbulence, alteration of the forest structure, and increased tree mortality [19,61].

### 2.3.2. Spatial Analysis

For the spatial analysis, we used a regular pixel grid with 10 km spatial resolution. Results from the forest fragmentation classification were quantified as the area (square kilometer) for each pixel annually between 1985 and 2017. In addition, we calculated the total number of active fires annually for each regular pixel in the grid, considering the 2012–2017 period.

### 2.3.3. Statistical Analyses

All statistical analyses were performed using R software [62], including regression and trend analysis. For regression, several models (linear and non-linear) were performed to identify the best model that explains the relationship between the monthly active fires, rainfall, and surface temperature.

For the trend analysis, we used two robust non-parametric methods that are not particularly sensitive to discrepant data, the Mann–Kendall test [63,64] and the Sen's Slope estimator [65] through the "wq" package in R [66]. The Mann–Kendall test was used to assess whether there is an upward or downward trend over time, while the Sen's Slope estimator was used to estimate the magnitude of the trends. We adopted a 5% significance level ($p \leq 0.05$) for all analyses.

## 3. Results

### *3.1. Forest Fragmentation Dynamics between 1985 and 2017*

Forest cover in the study area decreased by 35%, from 88,195 km$^2$ in 1985 to 56,893 km$^2$ in 2017 (Figure 2a), showing a net reduction of 31,302 km$^2$. Although the growth of secondary forests slightly increased total forest cover in 1990–1992, 2005, 2013, and 2015, a trend of significant reduction ($p < 0.05$) in forest cover was identified for the whole study period (Figure 2a). The reduction was equivalent to an average of 1078 km$^2$ year$^{-1}$ (Kendall's tau = −0.936). More than half of the forest cover during this period (1985–2017) corresponded to forest edges (average of 65% ± 3% year$^{-1}$). Around 21% ± 4% year$^{-1}$ corresponded to core areas, 12% ± 5% year$^{-1}$ corresponded to islets, and 2% ± 1% year$^{-1}$ corresponded to perforations (Figure 2b).

Although ongoing deforestation has induced overall landscape fragmentation of the study area, a close analysis of each fragmentation class reveals different trajectories during the study period (Figure 2c–f). The total area of core forest was reduced from 26,877 km$^2$ in 1985 to 10,029 km$^2$ in 2017 (63% of reduction), with a significant negative trend of 442 km$^2$ year$^{-1}$ ($p < 0.05$ and Kendall's tau = −0.830). The area of the forest edge was reduced from 54,752 km$^2$ in 1985 to 34,047 km$^2$ in 2017 (38% of reduction), with a significant negative trend of 805 km$^2$ year$^{-1}$ ($p < 0.05$ and Kendall's tau = −0.856). The area of perforations was reduced from 3111 km$^2$ in 1985 to 1103 km$^2$ in 2017 (65% of reduction), with a significant negative trend of 48 km$^2$ year$^{-1}$ ($p < 0.05$ and Kendall's tau = −0.689). On the other hand, the area of forest islets increased from 3455 km$^2$ in 1985 to 11,715 km$^2$ in 2017 (239% of increase), with a significant positive trend of 270 km$^2$ year$^{-1}$ ($p < 0.05$ and Kendall's tau = −0.936).

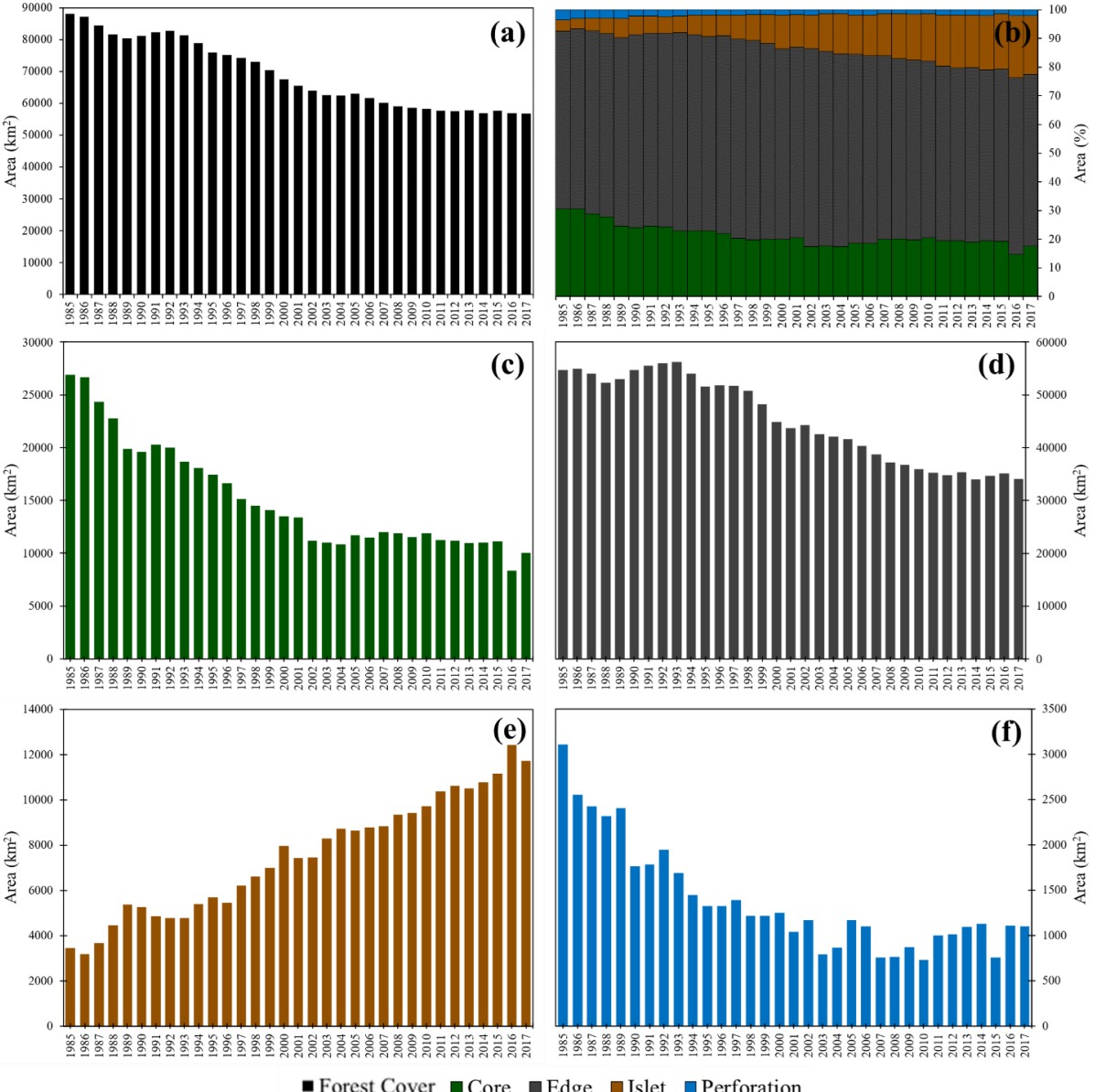

**Figure 2.** (**a**) Annual forest cover between 1985 and 2017 in the Maranhão Amazon, Brazil. (**b**) Fragmentation class during the study period as a percentage of the total forest cover in each year. (**c–f**) Annual area of each fragmentation class between 1985 and 2017. Note that the figures have the *y*-axis at different scales.

The pixel-by-pixel analysis (Figure 3) yielded general trends similar to those listed above. This spatial configuration also revealed the different temporal trajectories of the four fragmentation classes. Core areas showed significant trends ($p < 0.05$), with some pixels showing reductions of up to 0.360 km$^2$ year$^{-1}$ pixel$^{-1}$, and a few showing increases in the core forest of up to 0.197 km$^2$ year$^{-1}$ pixel$^{-1}$. For this fragmentation class, 95% of the trends indicated reductions, and 5% showed increases. Pixels with the largest decreases in the core fragmentation class appeared to be concentrated just outside or at the borders of Conservation Units (CUs) and Indigenous Lands (ILs). The edge fragmentation class also exhibited significant trends ($p < 0.05$), with around 79% of the areas showing reductions (up to 0.361 km$^2$ year$^{-1}$ pixel$^{-1}$), and 21% depicting increases (up to 0.337 km$^2$ year$^{-1}$ pixel$^{-1}$). Pixels with increasing trends in the edge class were mainly concentrated in the northern (coastal) region and areas surrounding the CUs and ILs. Pixels showing reduction trends appeared widespread throughout the study area, except within the CUs and ILs. The islet

fragmentation class also showed significant trends ($p < 0.05$), with only around 19% of the areas showing reductions of this fragmentation class (up to 0.198 km$^2$ year$^{-1}$ pixel$^{-1}$), and 81% showing increases (up to 0.276 km$^2$ year$^{-1}$ pixel$^{-1}$). Pixels showing increasing trends in the islet class were widespread throughout the region, except for the CUs and ILs. Pixels with reduction trends in this class were concentrated in the central and southern portions of the study area. Finally, the perforation fragmentation class also showed significant trends ($p < 0.05$), with around 54% of the areas showing decreasing trends (up to 0.198 km$^2$ year$^{-1}$ pixel$^{-1}$), and 46% showing increasing trends (up to 0.167 km$^2$ year$^{-1}$ pixel$^{-1}$). Most pixels with a higher tendency to both increase and decrease were found within CUs and ILs.

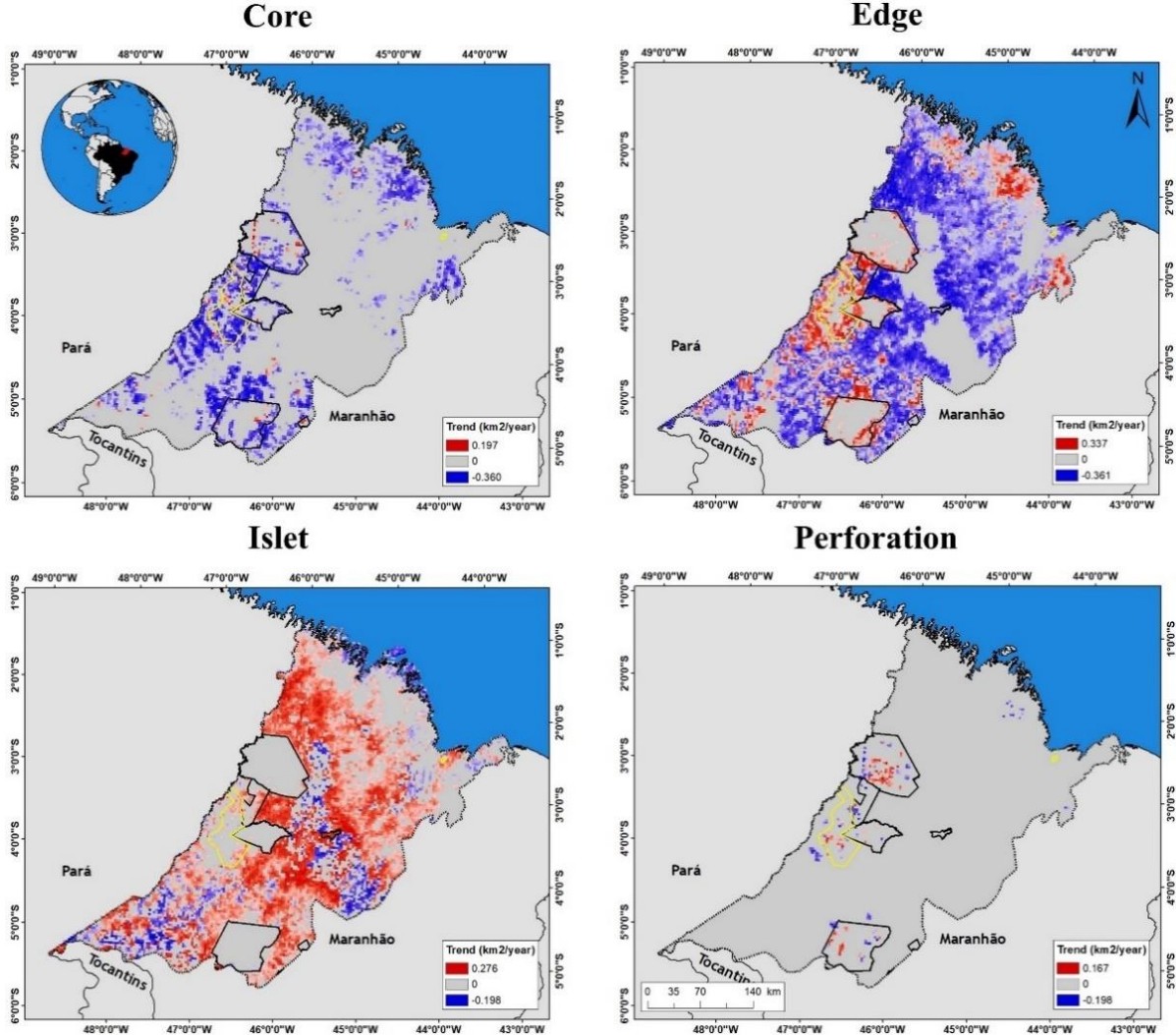

**Figure 3.** Spatial trends (Sen's Slope estimator) for each fragmentation class between 1985 and 2017 in the Amazon region of Maranhão, Brazil. Areas were calculated in pixels with 10 km spatial resolution. Negative values (in blue) represent the decrease trend, while positive values (in red) represent the increase trend.

### 3.2. Active Fire Dynamics between 2012 and 2017

Around 262,205 active fires were recorded in the study area between 2012 and 2017, with an annual average of 43,701 ± 21,763 fires year$^{-1}$. The forest fires showed a well-defined temporal pattern (Figure 4a), with a lower fire activity between January and July (wet season), and a higher activity between August and December (dry season). The lowest average number of active fires within forest areas was observed in March (8 ± 3 fires month$^{-1}$), while the maximum number of active fires was observed in December

(7024 ± 7533 fires month$^{-1}$). Fires on forest cover areas accounted for more than half (53 ± 7%) of all active fires registered in the region (Figure 4b) during the studied period, and 2015 was the year with the highest number of registered fires within forest cover areas (65% of the active fires registered that year).

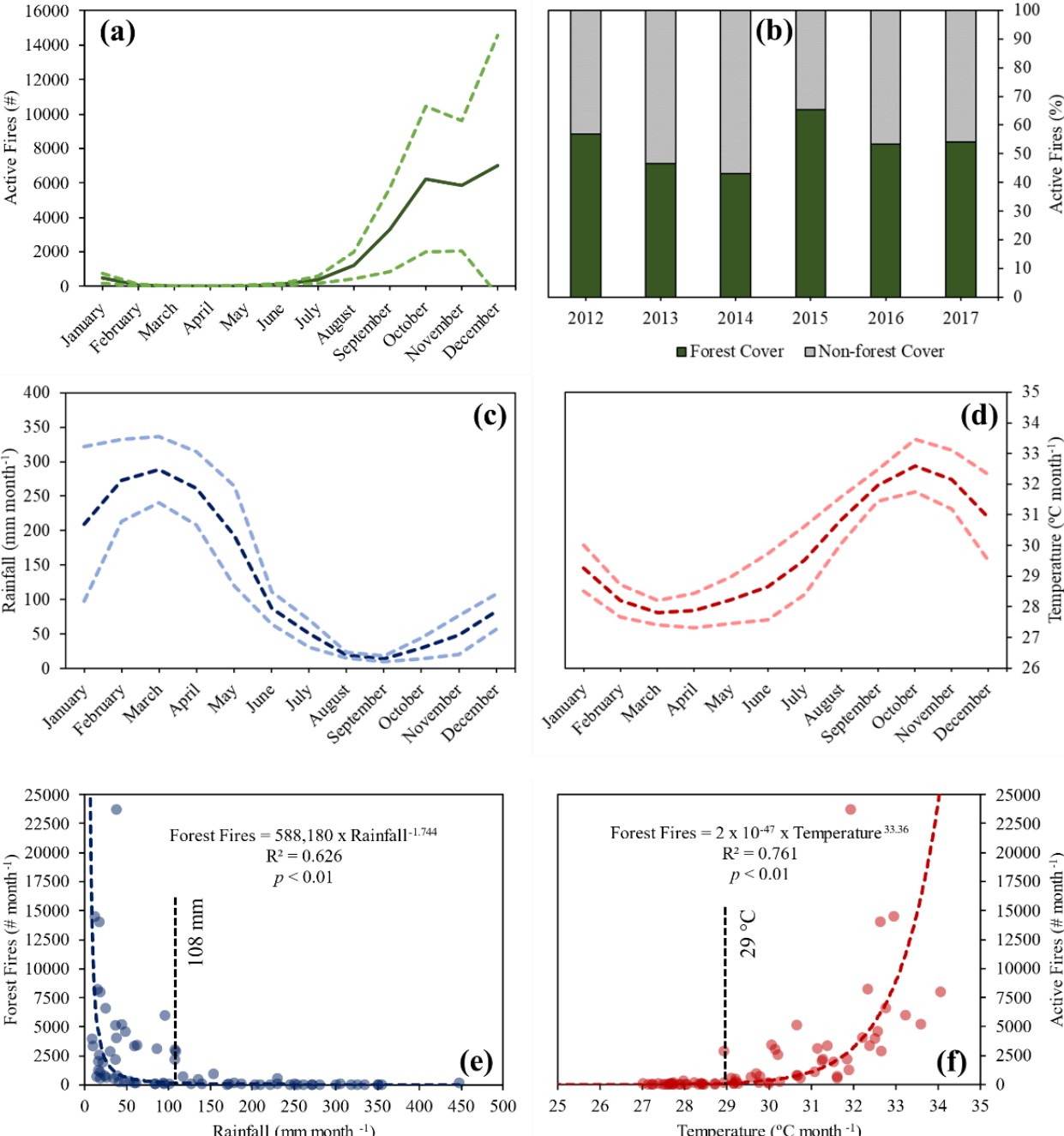

**Figure 4.** (**a**) Average monthly pattern of active fires in forest cover areas for the 2012–2017 period in the Amazon region of Maranhão, Brazil. (**b**) The proportion of active fires registered annually in forest and non-forested cover during the 2012–2017 period. (**c**) The monthly average rainfall in the 2012–2017 period. (**d**) The monthly average temperature in the 2012–2017 period. (**e**) Regression of the relationship between the monthly average rainfall and the monthly total number of active fires in forest cover in the 2012–2017 period. (**f**) Regression of the relationship between the monthly average temperature and the monthly total number of active fires in forest cover in the 2012–2017 period. The dashed lines in lighter colors in Figure 4a,c,d represent the standard deviation for the 2012–2017 period.

The forest's susceptibility to fire is associated with the rainfall seasonality (Figure 4c) and temperature (Figure 4d). These two variables combine to produce environmental conditions conducive to fire spread—the lowest fire activity within forest cover areas corresponding to the greater amount of rainfall and lowest temperature levels. However, a three-month delay was observed between the minimum monthly rainfall (15 ± 4 mm in September) and the maximum monthly record of active fires in forests (December); on the other hand, the maximum temperature record (33 ± 0.85 °C in October) preceded by two months the maximum record of forest fires (in December).

Regression analyses, shown in Figure 4e,f, showed that the best-adjusted model was a power function, with a strong and significant ($p < 0.01$) correlation between total monthly active fires and average rainfall ($R^2 = 0.626$) and temperature ($R^2 = 0.761$). Thus, while active fires progressively decreased with increasing rainfall, active fires increased with increasing average temperature. In addition, 2 thresholds were identified: first, the number of active fires remained below 1000 per month when monthly rainfall amounts stayed above 108 mm. Second, when the air temperature rose above 29°C, the number of active fires increased to monthly accumulated totals of over 1000.

Figure 5 shows the spatial distribution of annual active fires (for the period 2012–2017). The average number of annual active fires in forested land ranged from 1 to 38 active fires per pixel (10 km of spatial resolution), with 60% of the pixels having up to 2 active fires per year, and 2% of the pixels having between 19 and 38 active fires per year (Figure 5a). The pixels with higher hotspot annual averages (19 to 38) were concentrated mainly in the south-central region of the Maranhão Amazon, especially in the surroundings and interior of the IL Araribóia and the CU Gurupi Biological Reserve (Figure 5a). Figure 5b shows the spatial distribution of the standard deviation for the annual average of active fires for the 2012–2017 period.

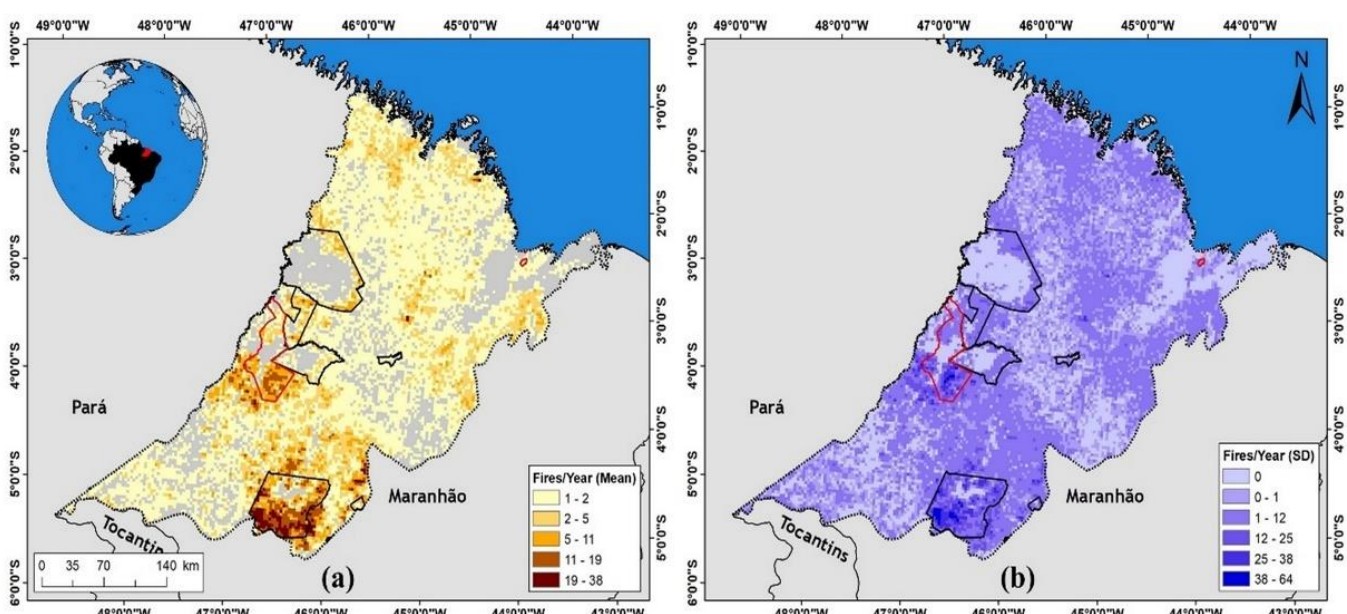

**Figure 5.** (**a**) Hot spot annual average for the 2012–2017 period in the Amazon region of Maranhão, Brazil. (**b**) Hot spot standard deviation for the period 2012–2017. We quantified active fires in pixels with 10-km spatial resolution.

### 3.3. Forest Fragmentation and Fires between 2003–2017

We used the burned areas to assess the extent of fires on forest cover, and explore its relationship with the forest fragmentation classes. Between 2003 and 2017, a total area of 25,201 km$^2$, or 1680 ± 1132 km$^2$ year$^{-1}$, was impacted by fire in the region, with a minimum burned area of 407 km$^2$ recorded in 2013, and a maximum of 4526 km$^2$ in 2015;

in addition, no significant trend in the burned area ($p > 0.05$ and Kendall's tau = $-0.086$) was observed during the study period (Figure 6).

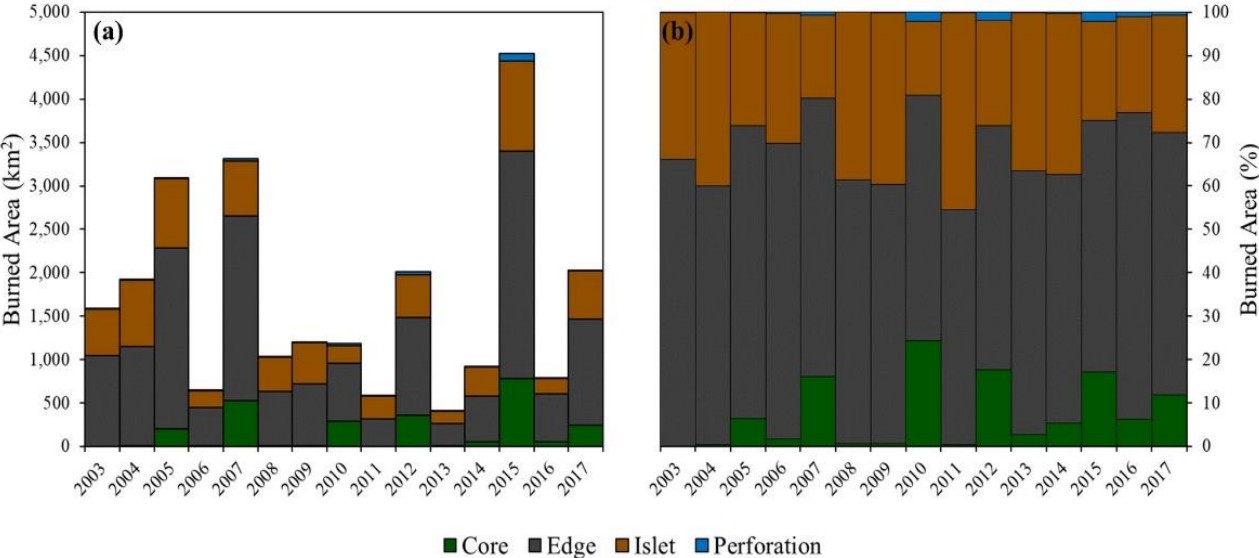

**Figure 6.** (**a**) Area of burnt forest per year in each fragmentation class between 2003 and 2017 in the Amazon region of Maranhão, Brazil. (**b**) The proportion of burnt forest per year in each fragmentation class between 2003 and 2017.

Areas in the edge fragmentation class experienced the greatest extent of burned areas. An average of around $1031 \pm 695$ km$^2$ on forest edges burned per year in the region between 2003 and 2017, the equivalent to $61 \pm 5\%$ of the total burned forest cover in the study. The second fragmentation class with the greatest extent of forest fires was the islet class, with an annual average of $466 \pm 253$ km$^2$ year$^{-1}$ burned between 2003 and 2017, the equivalent to $31 \pm 8\%$ of the total burned forests. An average of around $169 \pm 227$ km$^2$ of the forests core class burned annually during the study period (about $7 \pm 8\%$), while only $13 \pm 24$ km$^2$ (about $1 \pm 1\%$ of the total) of the perforation class burned per year. During the studied period, 2015 was the year with the greatest overall burned forest extent, and the year where each fragmentation class experienced its highest burned area extent (Figure 6a). Results from the trend analysis showed no statistically significant trend ($p > 0.05$) in burnt areas during the period 2003–2017 (Kendall's tau: core = 0.276; edges = $-0.086$; islands = $-0.181$; and perforation = 0.276).

## 4. Discussion

Here, we presented the first long-term analysis of the annual extent of forest fragmentation in Maranhão state's remaining Amazonian forests. This region was exposed to decades of predatory deforestation, which led to a 63% reduction in forest core areas between 1985 and 2017. We also highlighted the remarkable effect of this loss of core forest remnants by an astounding 239% increase in the area of islets (isolated areas of forest too small or too narrow to contain core forests). Vedovato et al. [10] reported that the forest islets in the Maranhão Amazon were 10,877.5 km$^2$ in 2014. However, our estimates for 2017 (11,715 km$^2$) show that the forest fragmentation continues to increase in Maranhão.

Another worrying trend observed in our study was that areas in the perforation class (edges inside core areas) were concentrated in the few legally protected forest remnants, likely due to illegal logging activities [8]. By extending deeply into previously intact forest areas, these fragments expose core forests to detrimental edge effects, including shifts in plant and animal community composition and changes in diversity [67,68], increased rates of tree mortality [69] and fire susceptibility [13,15,18,70], altered microclimates [71], and increased carbon emissions [14,72,73].

The high level of forest fragmentation revealed by our results showed that the secondary forest's growth did not offset the deforestation induced-fragmentation of old-growth forests in this region. In the Brazilian Amazon, deforestation of secondary forests surpasses old-growth forests, mainly because the secondary forests receive less protection than old-growth forests [74]. In Maranhão specifically, secondary forests have high annual rates of deforestation [9]. They are threatened by recent state land-use legislation [9,75], leading to the large-scale loss of these forests that provide essential ecosystem services [76–78]. Evidence in the Amazon has shown that large-scale deforestation has reduced the amount of rainfall [79,80] and affected agricultural production [81].

The recent weakening of deforestation enforcement and dismantling of environmental policies during the last three years in Brazil has led to uptrend deforestation rates in the Brazilian Amazon [82]. In addition to increasing fires directly associated with deforestation [26,29,83], this uptrend induces more forest fragmentation in this region, thereby compromising the achievement of sustainable development goals and international climate agreements.

Our findings indicate a noticeable seasonal pattern of forest fires in the Maranhão Amazon region. The analyses of active fires within forest cover show that forest fires in the Maranhão Amazon occurred mainly during the second half of the year, with a peak in December, which differs from the pattern observed elsewhere in the Brazilian Amazon, where peaks usually occur in August and September [84,85]. However, as Alencar et al. [84] observed for other Amazonian regions, the peak of forest fires in the Maranhão Amazon occurred shortly after the onset of the dry season, at the same time as the temperature increases in the region.

In the Amazon, fires have typically been linked to the deforestation process, resulting from establishing agricultural and pasture areas [29,30]. Fires from cleaning newly deforested areas or from agricultural and pasture areas often accidentally escape to adjacent forests through the edges [13,15,16]. Furthermore, the combination of these ignition sources from human actions [29,30], increased forest edges [13,15,16], and extreme droughts lead to an abnormal increase of forest fires [27,28]. During years of extreme drought, forests become hotter and drier, thus more likely to sustain uncontrolled fires [27,28,30]. In 2015, a drought year for the Brazilian Amazon [27,28], the Maranhão Amazon experienced the largest burned forests extent since 2007. In this region, in addition to the accidental escape of fire into the forests, criminal ignition sources by illegal loggers are common, mainly in protected areas and indigenous lands [6,8,9]. Forest fires also lead to negative socio-economic impacts, such as respiratory illness [31,32,86] in the Amazonian population, as well as economic losses [87].

Here, we showed that the forest fragmentation of the Amazonian forests of Maranhão is an essential factor that determines the occurrence of forest fires in the region. Almost all forest fires in the region between 2003–2017 occurred at the edge and islet fragmentation classes. Due to the alteration of forest remnant configuration in the landscape, forest edges and small forest patches (composed by edges only) have more supportive conditions for the spread of fire, including a drier and hotter microclimate, abundant availability of fuel (necromass due to tree mortality at the edges [72,88]) in the understory, and greater exposure to ignition sources, since they are surrounded by agricultural and livestock areas [13,15,17–20].

We showed that during extreme drought years, forest core areas (regions less exposed to human activities) were impacted by fires; this occurred because, during extreme drought years, the Amazonian forests becomes drier and hotter, making them more susceptible to spreading fire [27,28]. Our results confirmed that the Amazonian forests of Maranhão have become more fragmented and flammable [84,89]. The increase in the frequency, extent, and intensity of extreme droughts in the region during the 21st century, previously predicted by models [1,90,91], has been observed in the last two decades [27,28,58,92–94]. However, the warming of the Pacific and Atlantic surface ocean induced extreme droughts in the Amazon [27,92,93]. This phenomenon is predictable, making it possible to efficiently

produce a fire management plan during these years to avoid the spread of wild forests fires [95,96].

Finally, for REDD+ programs [97] in the Maranhão state, decision-makers should not just focus only on deforestation [98], but also on reducing emissions from forest degradation, including forest fires [27,99] and edge effect disturbances [14,73]. In addition, they should increase forest carbon stocks by protecting secondary forest areas that already exist in the state [9,74,76,78,100–102].

## 5. Conclusions

Here, we conclude that predatory exploitation of forest resources in the Maranhão Amazon has resulted in highly fragmented and fire-prone forests remnants. Thus, it is urgent to protect Maranhão forests from illegal deforestation and degradation (from edge effects and fires disturbances). Therefore, environmental and land policies elaborating with cooperation between the Maranhão state, civil society, the private sector, and scientists is necessary and urgent to protect old-growth and secondary forests in the region. On the other hand, in the context of the United Nations Decade on Ecosystem Restoration, this threatened area of the Brazilian Amazon is recognized as a forest restoration hotspot, and the protection of secondary forests is urgently needed to offset, in part, the historical forest fragmentation in the region.

Our work shows the already known diversity of fire types in the eastern region of the Brazilian Amazon. Furthermore, our findings reveal that forest fires depend not only on the climate, but also on the synergic interaction between the configuration of forest remnants and the availability of man-made ignition sources in the landscape. However, during extreme drought years, fire control measures are necessary to prevent the spread of uncontrolled forest fires.

Finally, our findings contribute to filling gaps in the understanding of forest fragmentation and its relationship with forest fires in the Amazon region of the state of Maranhão. Furthermore, future research should analyze the period after 2017 to understand forest fragmentation and fires during the recent increasing deforestation rates in the Brazilian Amazon.

**Author Contributions:** Data curation, C.H.L.S.-J.; Formal analysis, C.H.L.S.-J.; Methodology, C.H.L.S.-J.; Writing—original draft, C.H.L.S.-J., A.T.M.B. and D.S.B.; Writing—review and editing, O.S.C.J., A.L.S., L.O.D.B., A.L.S.S., S.T.A., C.T.A., A.T.G.F., G.X.R., D.C., F.B.S., M.S.S.P., S.A., M.K., L.B.V., L.O.A. and L.E.O.C.A. All authors have read and agreed to the published version of the manuscript.

**Funding:** This research was funded by Fundação de Amparo à Pesquisa e ao Desenvolvimento Científico e Tecnológico do Maranhão—FAPEMA, grant number 00813/19 and 02989/20. This study was financed in part by the Coordenação de Aperfeiçoamento de Pessoal de Nível Superior—Brasil (CAPES)—Finance Code 001. The University of Manchester funded this work through the project entitled "Forest fragmentation mapping of Amazon and its vulnerable margin Amazon-Cerrado transition forests". Part of this work was carried out at the Jet Propulsion Laboratory, California Institute of Technology, under a contract with the National Aeronautics and Space Administration (NASA). C. T. A was funded by Fundação de Amparo à Pesquisa do Estado de São Paulo (FAPESP) [grant number 2020/06734-0]. L.B.V thanks Coordenação de Aperfeiçoamento de Pessoal de Nível Superior—Brasil (CAPES) (number 88881.128127/2016-01). S.T.A. is currently receiving postdoctoral support from Programa de Fixação de Doutor from the Universidade Estadual de Maranhão (UEMA). L.O.A acknowledges MAP-FIRE (IAI—SGP-HW016), National Council for Scientific and Technological Development—CNPq (314473/2020-3 and 409531/2021-9), and São Paulo Research Foundation—(FAPESP 2016/02018-2 and 2020/16457-3). L.E.O.C.A. acknowledges São Paulo Research Foundation—(FAPESP-SHELL 2020/15230-5), CNPq (314416/2020-0) and Brazilian Space Agency (AEB) for the support.

**Institutional Review Board Statement:** Not applicable.

**Informed Consent Statement:** Not applicable.

**Data Availability Statement:** The data that support the findings of this study are all publicly available from their sources. Processed data, products, and code produced in this study are available from the corresponding author upon reasonable request.

**Conflicts of Interest:** The authors declare no conflict of interest. The funders had no role in the design of the study; in the collection, analyses, or interpretation of data; in the writing of the manuscript, or in the decision to publish the results.

**Appendix A**

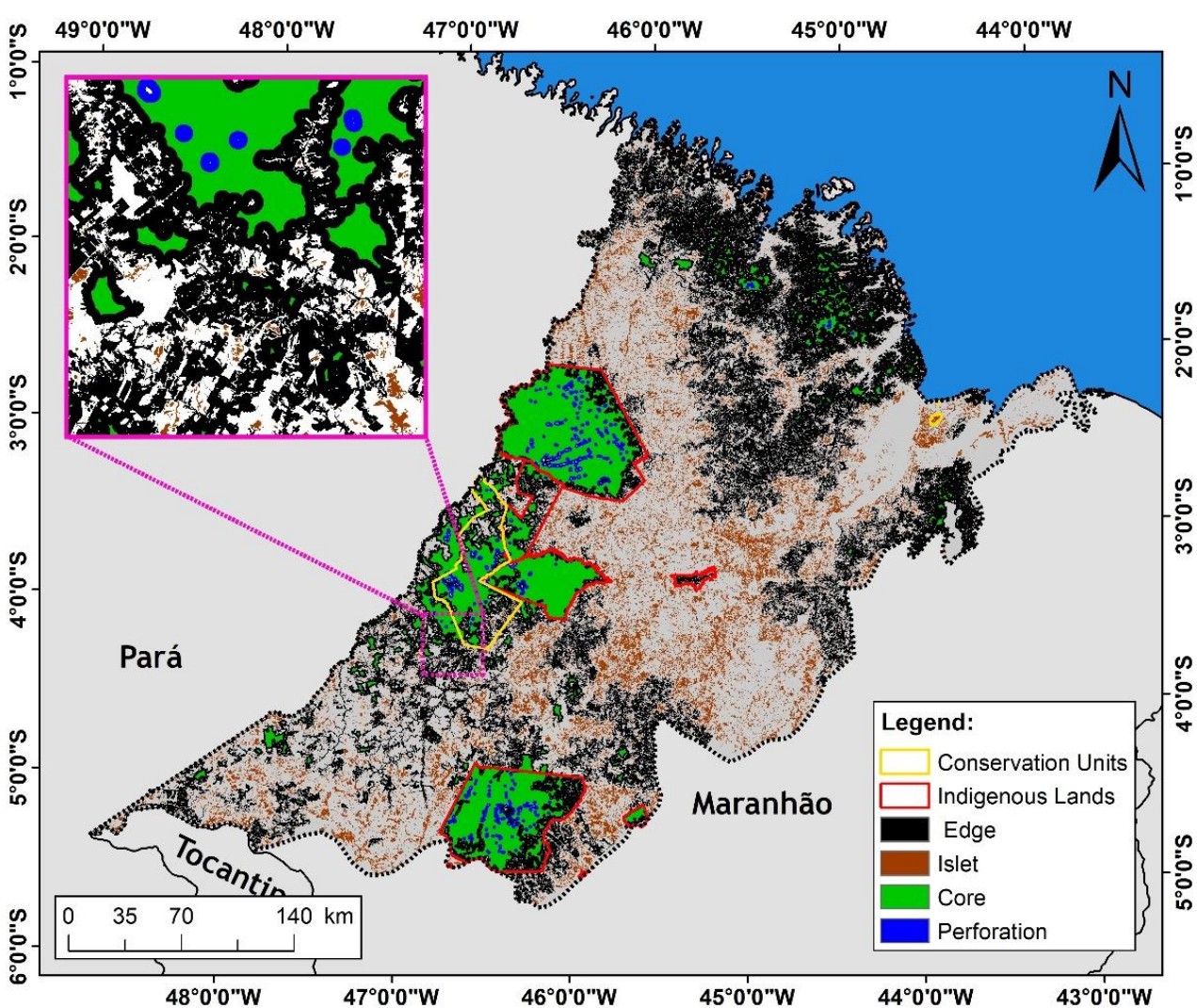

**Figure A1.** The result of the MSPA (Morphological Spatial Pattern Analysis) algorithm's classification using the 2017 forest map (see Figure 1).

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
