# Peer review of "Forest Fragmentation and Fires in the Eastern Brazilian Amazon–Maranhão State, Brazil"

_fire, doi:10.3390/fire5030077_

Round 1

Reviewer 1 Report

The topic of submitted article is relevant for Fire journal. Manuscript is well-written in the context of English language. But article has a set of disadvantages that must be processed within revision.

Introduction and Background are united in Introduction Section, but have a limited number of cited works. There are many works were cited in Discussion and Conclusion. I suggest to extend Introduction section using references from Discussion and Conclusion. After this, cited works should be used in Discussion section to compare and contrast obtained results through the references from Introduction. Conclusion should be completely reworked without references. Generalization of key findings and further considerations should be presented in Conclusion section.

Next, In materials and Methods section authors should provide short description of MSPA classification algorithms. It will be useful to compare suggested algorithm with classical classification algorithms like DBSCAN, k-means or hierarchical algorithms, at least using information from supporting references.

Then, In materials and methods section it is necessary to provide regression equations with descriptions of variables.

I suggest major revision.

Author Response

RESPONSES TO REVIEWER 1

Below, we repeat all Reviewers’ comments and reply to the concerns one by one. Each comment is in continuous sequence and with our responses in bold.

The topic of submitted article is relevant for Fire journal. Manuscript is well-written in the context of English language. But article has a set of disadvantages that must be processed within revision.

R: We are glad that the Reviewer liked the narrative and message of the paper. We thank the Reviewer for the comprehensive revision of the manuscript. We have incorporated all suggestions into the revised manuscript. We are sure that the revisions based on the Reviewer's comments have strengthened the new version of the manuscript.

Introduction and Background are united in Introduction Section, but have a limited number of cited works. There are many works were cited in Discussion and Conclusion. I suggest to extend Introduction section using references from Discussion and Conclusion. After this, cited works should be used in Discussion section to compare and contrast obtained results through the references from Introduction. Conclusion should be completely reworked without references. Generalization of key findings and further considerations should be presented in Conclusion section.

R: Thanks for the comments. We revised the Introduction, Discussion, and Conclusions sessions according to the Reviewer's suggestions. Please see our revised manuscript.

Next, In materials and Methods section authors should provide short description of MSPA classification algorithms. It will be useful to compare suggested algorithm with classical classification algorithms like DBSCAN, k-means or hierarchical algorithms, at least using information from supporting references.

R: We revised our Material and Methods section as suggested by the Reviewer. See below:

The MSPA is a customized sequence of mathematical morphological operators targeted at describing the geometry and connectivity of the image components [50]. The MSPA’s approach is based on geometric concepts and can be applied at any scale. The foreground area of a binary image is divided into the following visually distinguished MSPA classes: Edge - the outer perimeter of the forest class (including the Loop, Corridor, and Branch classes), Perforation - edges inside core areas, Core – the forest core area, excluding the edges, and Islets - the forest sections with the area too small to contain a core. More details and illustrations were available in the MSPA Guide (https://forest.jrc.ec.europa.eu/en/activities/lpa/mspa). See an example of MSPA classification in Figure A1.

Thanks for the suggestion, but algorithms like DBSCAN, K-means, or hierarchical algorithms are not usual in landscape fragmentation studies like MSPA; moreover, comparisons with other algorithms are outside the main aim of our manuscript.

Then, In materials and methods section it is necessary to provide regression equations with descriptions of variables.

R: All equations and variables were provided in the manuscript.

Reviewer 2 Report

Thank you for the opportunity to review this work, it is in an interesting area of the world, and the authors have nicely outlined the forested study area.

I found very few areas that  require any serious rework, but would make the following suggestions for improvement of the paper:

  1. In the Background, it would be good to have a small bit of background as to the forest ecology, structure and forest management regimes undertaken in this region for some broader context of links to fire ecology (vegetation fuels and the like).
  2. On a similar note, it was not very clear how the forest ecology or management links to the types of fragmentation (core, edge, islet etc.). Perhaps in section 3.1 this could be better spelt out, as there is clear link in conclusion alluding to this in lines 366-368 around fuel and understorey in edges, but not how this differs/compares to the other 3 fragmentation classes.
  3. There is a note around the number of fires in section 3.2 and the extent of fires in 3.3, but I was hoping to get a better link from the results correlating the area burned, size of fire and fire dynamics - is it increasing over time, and in what areas? The graphs in Figure 6 should make this clearer, and are difficult to read and interpret.
  4. Also in relation to 3.2 and 3.3, what is the fire return period like? What does this mean for forest recovery and secondary vegetation effects?
  5. There are a few grammatical errors - what are famountires on line 248? I am unfamiliar with the word.
  6. Line 158 why is spatial analysis in italics?
  7. Figures 2 - could these have the % reduction overlaid as outlined in the text?
  8. Figure 2 a and b are quite dark

Author Response

RESPONSES TO REVIEWER 2

Below, we repeat all Reviewers’ comments and reply to the concerns one by one. Each comment is in continuous sequence and with our responses in bold.

I found very few areas that require any serious rework, but would make the following suggestions for improvement of the paper:

R: We are glad that the Reviewer liked the narrative and message of the paper. We thank the Reviewer for the comprehensive revision of the manuscript. We have incorporated all suggestions into the revised manuscript.

In the Background, it would be good to have a small bit of background as to the forest ecology, structure and forest management regimes undertaken in this region for some broader context of links to fire ecology (vegetation fuels and the like).

R: Forest ecology, structure, and forest management in this piece of the Brazilian Amazon within the state of Maranhão are essential points. However, even with its importance, this region was poorly studied in these aspects. Our group intends to propose research projects to fill these gaps in the coming years. In addition, we believe that our work to understand fragmentation and fire regimes in the region is an important basis for future work.

On a similar note, it was not very clear how the forest ecology or management links to the types of fragmentation (core, edge, islet etc.). Perhaps in section 3.1 this could be better spelt out, as there is clear link in conclusion alluding to this in lines 366-368 around fuel and understorey in edges, but not how this differs/compares to the other 3 fragmentation classes.

R: Thanks for this comment. We modified the paragraph on lines 366-368 to address the Reviewer's concern. Please see below:

Here, we showed that the forest fragmentation of the Amazonian forests of Maranhão is an essential factor that determines the occurrence of forest fires in the region. Almost all forest fires in the region between 2003-2017 occurred at edges and islet fragmentation classes. Due to alteration of forest remnant configuration in the landscape, forest edges and small forest patches (composed by edges only) have more supportive conditions for the spread of fire, including a drier and hotter microclimate, abundant availability of fuel (necromass due to tree mortality at edges [63,85]) in the understory and greater exposure to ignition sources since they are surrounded by agricultural and livestock areas [12–17].

There is a note around the number of fires in section 3.2 and the extent of fires in 3.3, but I was hoping to get a better link from the results correlating the area burned, size of fire and fire dynamics - is it increasing over time, and in what areas? The graphs in Figure 6 should make this clearer, and are difficult to read and interpret.

R: Relating the number of active fires and the burned areas is beyond the scope of our work. To make the two datasets more transparent, we have included an opening sentence in the description of the two products. Thus, the active fire data show fire activity (which will not necessarily lead to burnt areas); at the same time, the burnt area data reveal the actual extent of the areas impacted by the fire. Figure 6 does not have this aim.

Also in relation to 3.2 and 3.3, what is the fire return period like? What does this mean for forest recovery and secondary vegetation effects?

R: The fire return period, post-fire forest recovery, and the effects on secondary forests are the object of study of the group's ongoing projects, not being the aim of the present work.

There are a few grammatical errors - what are famountires on line 248? I am unfamiliar with the word.

R: Thanks, the typo has been fixed.

Line 158 why is spatial analysis in italics?

R: Thanks, the typo has been fixed.

Figures 2 - could these have the % reduction overlaid as outlined in the text?

R: The authors did not understand the reviewer's question.

Figure 2 a and b are quite dark

R: Figure 2 has been revised.

Reviewer 3 Report

Title: Forest fragmentation and fires in the eastern Brazilian Amazon – Maranhão state, Brazil

Overview:
This paper presents analysis of forest fragmentation and fires in Maranhão state, Brazil using a range of remotely sensed derived data including active fire, burned area, and climate variables. It is well written and seems an important study for understanding forest loss in this important region. I only have minor comments below. 

Other comments:

Line 26 I would “fire” instead of “fires” here.

Line 49 Should “util” be “until” or “in”?

Line 107 I would use “VIIRS active fire data” rather than “VIIRS active fires data”

Line 119 I see another use of “fires” instead of “fire”. I will stop pointing these out, but please check the rest of the manuscript. They are in the context of fire as an event/phenomenon rather than specifically meaning multiple fires (I hope that makes sense!)

Line 142 there is spacing issue around a )

Line 248 appears to have a typo: famountires 

Figure 6: Do variations in fire activity between years have any connection to temperature or precipitation variations between years? 

Lines 312-313 Should there be a sentence split in here? Currently reads …”which lead to a reduction of 63% of the forest loss occurred in forest core areas from 1985 and 2017…”

Line 321 should “extended” be “extending”?

Author Response

RESPONSES TO REVIEWER 3

Below, we repeat all Reviewers’ comments and reply to the concerns one by one. Each comment is in continuous sequence and with our responses in bold.

This paper presents analysis of forest fragmentation and fires in Maranhão state, Brazil using a range of remotely sensed derived data including active fire, burned area, and climate variables. It is well written and seems an important study for understanding forest loss in this important region. I only have minor comments below.

R: We are glad that the Reviewer liked the narrative and message of the paper. We thank the Reviewer for the comprehensive revision of the manuscript. We have incorporated all suggestions into the revised manuscript.

Line 26 I would “fire” instead of “fires” here.

R: Thanks, the typo has been fixed.

Line 49 Should “util” be “until” or “in”?

R: Thanks, the typo has been fixed.

Line 107 I would use “VIIRS active fire data” rather than “VIIRS active fires data”

R: Thanks, the typo has been fixed.

Line 119 I see another use of “fires” instead of “fire”. I will stop pointing these out, but please check the rest of the manuscript. They are in the context of fire as an event/phenomenon rather than specifically meaning multiple fires (I hope that makes sense!)

R: Thanks, we revised all the manuscript.

Line 142 there is spacing issue around a )

R: Thanks, the typo has been fixed.

Line 248 appears to have a typo: famountires

R: Thanks, the typo has been fixed.

Figure 6: Do variations in fire activity between years have any connection to temperature or precipitation variations between years?

R: Yes. As we found in our results and explained in the manuscript, during extreme drought years in the Amazon, the anomalous reduction in rainfall and the increase in temperature make forests more susceptible to the occurrence and spread of fire. In this context, fire activity can increase when there are anthropic sources of ignition.

Lines 312-313 Should there be a sentence split in here? Currently reads …”which lead to a reduction of 63% of the forest loss occurred in forest core areas from 1985 and 2017…”

R: Thanks, the sentence has been fixed in the manuscript.

Line 321 should “extended” be “extending”?

R: Thanks, the correction has been made.

Round 2

Reviewer 1 Report

The topic of submitted article is relevant for Fire journal. Manuscript is well-written in the context of English language.

Introduction and Background are united in Introduction Section, but have a limited number of cited works. During revision only five references were added to Introduction. There are many works were cited in Discussion and Conclusion. I suggest to extend Introduction section using references from Discussion and Conclusion. After this, cited works should be used in Discussion section to compare and contrast obtained results through the references from Introduction. Further considerations and future researches should be briefly presented in Conclusion section.

I suggest minor revision.

Author Response

We thank the Reviewer. We have incorporated all suggestions into the revised version of our manuscript.